# Probabilistic Risk Characterization of Heavy Metals in Peruvian Coffee: Implications of Variety, Region and Processing

**DOI:** 10.3390/foods12173254

**Published:** 2023-08-29

**Authors:** Grobert A. Guadalupe, Segundo G. Chavez, Erick Arellanos, Eva Doménech

**Affiliations:** 1Instituto de Investigación Para el Desarrollo Sustentable de Ceja de Selva (INDES-CES), Universidad Nacional Toribio Rodríguez de Mendoza de Amazonas, 342 Higos Urco, Chachapoyas 01001, Peru; segundo.quintana@untrm.edu.pe; 2Instituto de Investigación, Innovación y Desarrollo Para el Sector Agrario y Agroindustrial de la Región Amazonas (IIDAA), Universidad Nacional Toribio Rodríguez de Mendoza de Amazonas, Chachapoyas 01001, Peru; 3Instituto de Investigación en Ingeniería Ambiental (INAM), Universidad Nacional Toribio Rodríguez de Mendoza de Amazonas, Higos Urco 342, Chachapoyas 01001, Peru; erick.arellanos@untrm.edu.pe; 4Instituto Universitario de Ingeniería de Alimentos Food-UPV, Universitat Politècnica de València, Camino de Vera s/n, 46022 Valencia, Spain

**Keywords:** chemical contaminants, consumer exposure, probabilistic approach, risk assessment

## Abstract

Heavy metals are chemical contaminants, toxic, potentially carcinogenic and/or mutagenic, stable, persistent and are of concern in the food chain. The risk to the consumer of the presence of inorganic arsenic (iAs), cadmium (Cd), chromium (Cr), mercury (Hg) and lead (Pb) in five varieties (Bourbon, Típica, Catimor, Caturra and Pache) of parchment coffee from five regions (Amazonas, Cajamarca, Cusco, Huánuco and San Martín) was investigated in this study. A predictive model of the stages of coffee bean hulling, roasting and infusion was built to simulate the process. The results by region showed significant differences in which San Martín had the highest iAs, Cr and Pb values. The variety was only significant for Cr, of which Pache presented the highest concentration. The Cd and Hg values were below the detection limits. The hazard index (HI) was less than 1 for iAs, Cd, Cr and Hg and the combination of margin of exposure and the probability of exceedance (MOE-POE) for Pb indicated that an adverse health effect was not likely. The cancer risk (CR) for iAs and Pb in the 95th percentile was considered as both high and acceptable, respectively.

## 1. Introduction

Coffee is one of the most popular beverages worldwide, with an estimated consumption of 10 billion kg in 2020/2021 [1]. It is also one of the food products with the highest economic and social impact [2,3]. In Peru, the seventh largest exporter of this product worldwide and the second largest organic coffee exporter [4], coffee is the largest agricultural export and 223,482 smallholder families (≤5 hectares) are engaged in coffee production [5].

The botanical family of coffee plants has about 500 genera and more than 6000 species. From a commercial point of view, the most important genus is Coffea (Rubiaceae family) in which two species stand out: *Coffea arabica* (Arabica), which accounts for about 60% of world production and *Coffea canephora* (Robusta), which accounts for about 40% of traded coffee [6].

Culturing selected varieties that provide a balance between flavor, acidity, body and aroma in the cup is a crucial strategy in improving market competitiveness. The most widely cultivated varieties in Peru are Bourbon, Típica, Caturra, Pache and Catimor, all of which belong to the Arabica species [7,8]. Bourbon and Típica are varieties of excellent quality, although they are not highly productive. A natural mutation of Bourbon gave rise to the Caturra and Pache varieties, with similar levels of quality. Catimor is a blend of Caturra (Arabica) and Timor (Robusta) and is therefore low in acidity and slightly bitter in taste.

Coffee’s chemical composition is rich in potassium, niacin, magnesium, tocopherols, polysaccharides, monosaccharides, lipids, sterols and antioxidant substances such as polyphenols, chlorogenic acids, hydroxycinnamic acids, caffeine and caffeic acid [9,10,11]. Recent studies have linked moderate coffee consumption (2–5 cups a day) to preventing diseases of the circulatory, digestive and nervous systems [12,13]. However, these benefits can be compromised by the presence of contaminants such as heavy metal(oid)s, whose presence in the environment can have a natural origin (i.e., volcanic eruptions, leaching from soil and rocks, etc.) or may be caused by industrial and agricultural activities (inadequate waste management, biosolids, etc.) [14,15,16,17]. These stable and persistent elements accumulate in the soil, water and air and are considered to be worrying contaminants in the food chain [16,18,19]. Ninety percent of the population exposure is estimated to occur through contaminated food. Uptake and bioaccumulation of these substances can cause a broad array of toxic and mutagenic effects on target organs [18,20]. Inorganic arsenic (iAs), cadmium (Cd), chromium (Cr), lead (Pb) and mercury (Hg) can cause circulatory, nervous and enzymatic problems, as well such as lung cancer, skin cancer, bone fractures and kidney dysfunction [15,21,22,23,24].

As applied to chemical hazards, risk assessment is a systematic process by which potential hazards and consumer health effects are characterized over a given period [25,26]. A probabilistic approach is essential in this process to consider the input data’s variability to avoid overestimating the risk, so a probability distribution function (i.e., pdf) of consumption data, chemical contaminant concentration and weight difference between population groups is essential to reduce the risk uncertainty [27,28,29].

The metal content in a cup of coffee is directly related to the concentration of the contaminant in the raw beans and processing steps such as hulling, roasting and grinding, as well as the brewing method, i.e., the coffee/water ratio or brewing time [30]. Predictive modeling (PM), now established as a scientific discipline to support risk assessment, allows the formulation of stochastic processes and the evolution of the prediction of hazard as a function of possible intrinsic and extrinsic process parameters [31].

This study aimed to contribute knowledge regarding the lack of research on the concentration of metals in coffee from Peru, taking into account the variety, the region and the influence of the process, as well as providing a probabilistic view of the risk to which consumers are exposed by the consumption of Peruvian coffee. With this objective, a predictive model of the hulling, roasting and infusion stages was obtained. The concentrations of iAs, Pb, Cr, Cd and Hg were analyzed in five varieties of parchment coffee beans (Bourbon, Típica, Catimor, Caturra and Pache) harvested in the leading Peruvian coffee-growing regions (Amazonas, Cajamarca, Cusco, Huánuco and San Martín) and a probabilistic risk assessment and a sensitivity analysis of the variance were carried out.

## 2. Materials and Methods

### 2.1. Sample Collection

The stages in the coffee process we used for quantifying the risk are shown in Figure 1. A total of 9 samples of parchment beans from the province of Moyobamba (San Martín) were processed and analyzed to obtain the predictive models (PMs) of coffee hulling, roasting (210 °C for 10 min) and infusion, prepared in espresso, French and Italian coffee machines. First, 17 g of ground roasted beans was used to prepare a 50 mL cup of espresso coffee in a Ruby Pro espresso machine (water temperature 92 °C, pressure 1.3 bar and 30 s of percolation). For French coffee, 7.5 g of coffee and 120 mL of water at 92 °C were infused for 3 min, while Italian coffee was made from 35 g of coffee and 400 mL of water.

One hundred and fifty-nine samples of parchment bean coffee were collected between September and December 2021 to determine the concentration of iAs, Pb, Cr, Cd and Hg in a cup of coffee and to assess consumer risk. Beans were purchased directly from the coffee farms in the major producing regions: Amazonas (Rodríguez de Mendoza and Utcubamba), Cajamarca (San Ignacio), Cusco (La Convención and Calca), Huánuco (Leoncio Prado) and San Martín (Moyobamba, Rioja and Lamas), Figure 2. The varieties analyzed were Bourbon (n = 27), Típica (n = 36), Catimor (n = 51), Caturra (n = 39) and Pache (n = 6), all belonging to the Arabica species.

### 2.2. Chemical Analysis

The metals, iAs, Cd, Cr, Hg and Pb, were analyzed in the Soil and Water Research Laboratory (LABISAG) of the Universidad Nacional Toribio Rodríguez de Mendoza de Amazonas of Peru, accredited in accordance with ISO/IEC 17025:2017. The metal concentrations were determined following the method described in [32]. Thus, 1 g of ground parchment, green and roasted coffee beans and 2 mL of coffee brew were filtered, acidified and digested by an Agilent microwave plasma–atomic emission spectrometry (MP-AES), model 4100 MP-AES, with an inductively coupled plasma (ICP) method [33]. The spectral intensity was the mean of 3 repeated readings per sample. Detection wavelengths of 228.802 nm, 405.781 nm, 193.695 nm, 425.433 nm and 253.652 nm were selected to quantify iAs, Cd, Cr, Hg and Pb, respectively.

The equipment was calibrated by standard solutions of each element in different concentrations, prepared from a 1000 mg/kg standard solution. After each reading, the equipment recovered concentration and intensity without enriching the samples. The limit of detection (LOD) refers to the corresponding three times the value of the instrument background signal is generated by the matrix blank. The limit of quantification (LOQ) was determined by a signal-to-noise ratio of 10:1. Detection and quantification of iAs, Cd, Cr, Hg and Pb were controlled for analytical quality by measuring blind and double samples. The validation parameters were determined from ten replicates (see Table 1). The standard solutions used were of Agilent brand, located in the spectrometry area of LABISAG. After each reading, the equipment recovered both concentration and intensity without the need to enrich samples.

### 2.3. Risk Assessment

The risk for non-carcinogenic effects was found using the hazard quotient (HQ), Equation (1) [28], in which EDIxi is the estimated daily intake of each metal (*x*: iAs, Cd, Cr, Hg and Pb) and region (*i*: Amazonas, Cajamarca, Cusco, Huánuco and San Martín regions), obtained using Equation (2). RfD is the reference dose, whose values for iAs, Cd, Cr, and Hg were 3E-04, 1E-03, 1.5 and 1E-04 mg/kgBw/day, respectively [34].
(1)$${HQ}_{Xi}=\frac{EDIxi}{RfD}$$
(2)$$EDIxi\;(mg/kgBw/day)=\frac{\begin{array}{c} Cxi\left(\frac{mg}{kg}\right)*IR\left(\frac{kg}{day}\right) \end{array}}{Bw\left(kgBw\right)}$$

Cxi (mg/kg) is the concentration of metals in coffee samples from a region. In cases with values below the limit of detection (LOD), the LOD/2 concentration was considered. To account for the random uncertainty of the input data, metal concentrations were fitted to a probabilistic distribution function (pdf) in @Risk 8 software (Palisade, Newfield). IR (kg/day) is the ingestion rate, obtained by fitting the European consumption to a lognormal distribution defined by the 5th, 50th and 95th percentiles (5%, 24.1; 50%, 99.3; 95%, 409.1). Bw (kgBw) is the body weight reported by Quintanilla et al., (2019) [35] as a gamma distribution (5%, 53.95; 50%, 63.7; 95%, 83.91).

To assess the potential risk of adverse health effects from a mixture of chemical constituents, the hazard index (HI) was obtained by Equation (3):(3)$${HI}_{x}=\sum_{n=1}^{x}{HQ}_{n}$$

Conventionally, an HI less than 1 indicates that the total exposure does not exceed the level considered to be “acceptable”, and consumers are unlikely to be exposed to a toxic level with possible health consequences. On the other hand, if it exceeds 1 there is a possibility of suffering adverse effects [36,37].

For genotoxic effects, the margin of exposure (MOE), probability of exceedance (POE) and cancer risk (CR) processes were conducted to assess the carcinogenic risk of each metal *x*: iAs and Pb, per region i: Amazonas, Cajamarca, Cusco, Huánuco and San Martín (Equations (4)–(6)), respectively.
(4)$${MOE}_{xi}=\frac{{BMDL}_{\%x}}{{EDI}_{xi}}$$
(5)$${POE}_{xi}=Pr\left({EDI}_{xi}>{BMDL}_{x}\right)=\int_{{BMDL}_{\%x}}^{\infty }f(E)\;dE$$
(6)$${CR}_{xi}={EDI}_{xi}*{SF}_{x}$$
where EDI*_xi_* is the exposure obtained by Equation (2). BMDL_%_*_x_* is the lower confidence limit for an increase in percentage response for each *x* metal. The BMDL_01_ for iAs was considered a uniform distribution from 0.3E-03 to 8E-03 mg/kgBw/day for the effect lung, skin and bladder cancer, plus skin lesions [26,38,39,40]. For Pb and the effects of cardiovascular and nephrotoxicity, the reference values were BMDL_01_: 1.50E-03 mg/kgBw/day and BMDL_10_: 0.63E-03 mg/kgBw/day, respectively [38]. SFx is the slope factor, considered to be 1.5 (mg/kgBw/day)^−1^ for iAs and 8.5E-03 (mg/kgBw/day)^−1^ for Pb [34].

In the interpretation of the MOE results, for iAs, the Committee on Carcinogenicity of Chemicals in Food, Consumer Products and the Environment proposed that a MOE of 10 or higher associated with a 0.5% increased risk of lung cancer in humans could be considered of low concern [41]. According to the Swedish National Food Agency (SNFA), a value of 1–10 should be considered as a high-to-moderate risk, 10–100 as moderate-to-low risk and higher values as no risk [42]. Concerning Pb, a MOE of 10 or greater is a minor public health concern [43]. To interpret POE, a zero value implies that the reference value has not been exceeded in any case, meaning the risk is negligible [44]. For the cancer risk, values below 1E-06 are considered as a non-significant health risk; values between 1E-06 and 1E-04 are considered acceptable and values above 1E-04 mean the risk is high and indicate potential harm to humans [15].

The consumer health risk obtained by different formulations was quantified by simulation using a spreadsheet model in Microsoft Excel in @Risk 8 software (Palisade, Newfield). The propagation of metals along the chain was simulated by a standard Monte Carlo method with Latin hypercube sampling in 20 repetitions of 100,000 iterations per simulation in each scenario. Several sensitivity studies were also performed to determine the influence of the process stages from the parchment beans to the cup of coffee in risk characterization metrics.

### 2.4. Statistical Analysis

A multifactor analysis of variance (ANOVA) and Tukey’s mean comparison test with a significance level α = 0.05 were performed, in SPSS V.25 statistical software, (Chicago, IL, USA) to study the influence of the coffee variety and the production region in the concentration of heavy metals.

## 3. Results and Discussion

### 3.1. Predictive Modeling

The final concentration of heavy metals in coffee brews varies with the technological processes used such as hulling, roasting or brewing [45]. Figure 3 shows the iAs, Cr and Pb concentrations (metals with values above the LOD) in parchment, green and roasted beans and in the infusions obtained from the espresso, French and Italian coffee machines. These metals and their mean concentrations and deviation in parchment beans were iAs (1.06 ± 0.1 mg/kg), Cr (0.39 ± 0.07 mg/kg) and Pb (0.72 ± 0.04 mg/kg); in green beans, iAs (0.75 ±0.02 mg/kg), Cr (0.08 ± 0.005 mg/kg) and Pb (0.34 ± 0.10 mg/kg); in roasted beans iAs (0.68 ± 0.05mg/kg), Cr (0.08 ± 0.004 mg/kg) and Pb (0.29 ± 0.01 mg/kg). In coffee beverages, the mean values of iAs in the espresso, French and Italian machines were 0.041 ± 0.006, 0.040 ± 0.004, 0.42 ± 0.005 mg/kg, in Cr 0.01 ± 0.006, 0.01 ± 0.002, 0.01 ± 0.005 mg/kg and in Pb 0.03 ± 0.01, 0.04 ± 0.01, 0.02 ± 0.01 mg/kg, respectively. The differences between the coffee machines were not significant for iAs and Cr (*p*-value 0.5413; 0.0807), but they were for Pb (*p*-value 0.0044).

Previous studies on green coffee gave very different values for the metals studied. The iAs concentration ranged between the LOD and 0.1 mg/kg [6,46,47]. Cr was not detected in the studies such as those by Berego et al. (2023) [48]; Getachew and Worku (2014) [49]; Şemen et al. (2017) [6]. Values around E-03 mg/kg were reported by Dubale et al. (2017) [50] in Ethiopian coffees and Rodrígues et al. (2011) [51] in Hawaiian beans. Our values of around E-02 mg/kg were comparable to those of Várady et al. (2021) [52] (0.06 mg/kg) and Santato et al. (2012) [46] (0.017–0.087 mg/kg). Values close to E-01 mg/kg were obtained by Árvay et al. (2019) [53] (0.12 ± 0.04 mg/kg) and Gure et al. (2017) (0.28 ± 0.01 mg/kg). Albals et al. (2021) [54] published higher Cr values obtained from beans bought in the Jordanian market (2.26 ± 0.54 mg/kg). Samples collected from seven countries on three continents were analyzed by Vezzulli et al. (2023) [47]; the Cr results showed substantial variability (<LOQ–4.2mg/kg). Our Pb results were in line with those obtained by Semen et al. (2017) [6], in Turkish coffee beans, whose values ranged from 0.06 to 0.3mg/kg. These findings were lower than those published by Albals et al. (2021) [54] (2.45 ± 1.79) and Nogaim et al. (2014) [55], who obtained a mean value of 2.07 mg/kg, a minimum of 0.6 and a maximum of 7.99 mg/kg. Lower values were reported by Adler et al. (2019) [45] (0.076 ± 0.09 mg/kg); Jeszka-Skowron et al. (2016) [56] (0.025 mg/kg) and Vezzulli et al. (2023) [47] (>LOQ to 0.4 mg/kg). Pb concentration was below the LOQ in the studies by Ashu and Chandravanshi (2011) [14], Getachew and Worku (2014) [49], Dubale et al., (2017) [50] and Berego et al. (2023) [48]. Cd and Hg were not detected in our study, which coincided with the findings of Ashu and Chandravanshi (2011) [14], Getachew and Worku (2014) [49]; Dubale et al., (2017) [50] and Berego et al., (2023) [48].

Kowalska (2021) [15] found no iAs in roasted coffee. The published Cr values have generally been low (<LOD–0.1 mg/kg), such as those of Getachew and Worku (2014) [49], Árvay et al. (2019) [53] and Massoud et al., (2022) [57]. Wide variability was found in the results obtained by Mohammed et al. (2019) [58] (<LOD–7.5 mg/kg) and Pigozzi et al. (2018) [16] (<LOD–3.27 mg/kg). Finally, our Pb results are in the range of those published by Kowalska (2021) [15] (0.021–0.791 mg/kg) and Pigozzi et al. (2018) [16] (0.14–2.59 mg/kg). However, they were higher than studies that did not detect this element, such as those by Ashu and Chandravanshi (2011) [14]; Gebretsadik et al. (2015) [59]; Getachew and Worku (2014) [49] and Massoud et al. (2022) [57]. Higher values (between 0.7 and 7 mg/kg) were published by Albals et al. (2021) [54] (3.01 mg/kg); Da Silva et al. (2017) [60] (0.75 ± 0.33 mg/kg); Mohammed et al. (2019) [58] (0.9–3.9) and Da Silva et al. (2017) (0.75 mg/kg). No Cd or Hg were detected in the present study, as was also found by Ashu and Chandravanshi (2011) [14], Getachew and Worku (2014) [49], Gebretsadik et al. (2015) [59], Kowalska (2021) [15] and Massoud et al. (2022) [57]. However, values of Cd in the order of E-02 mg/kg were reported by Suseela et al. (2001) [61], Da Silva et al. (2017) [60], Adler et al. (2019) [45] and Árvay et al. (2019) [53]. Higher values, around E-01, were published by Albals et al. (2021) [54] (0.45 mg/kg) and Berego et al. (2023) [48] (0.80 ± 2.52 mg/kg) regarding market samples from Ethiopia.

The Cd and Pb results in infused coffee coincided those of with other authors whose values were very low or not detected [14,59,60]. Higher values of Pb (0.133–0.558 mg/kg) were observed by Adler et al. (2019) [45] and even higher values of Pb were published by Nędzarek et al. (2013) [62] (0.62–1 0.24 mg/kg). The Cr results were in line with those reported by Da Silva et al. (2017) [60] (<LOD–0.0025); Janda et al. (2020) [63] (0.037 mg/kg) and Nędzarek et al. (2013) [62] (0.028–0.12) and were lower than those of Onianwa et al. [64] (1999) (0.89 ± 6.98 mg/kg).

To obtain the PM between stages, i.e., reduction in hulling, roasting and infusion, data were fitted with the @Risk program, obtaining a probability distribution function (pdf) for each metal (see Table 2).

The results showed that Cr was the metal with the highest reduction in the hulling stage, followed by Pb and iAs. The endosperm covering the beans is removed during roasting. The results showed that the reduction of metals after roasting did not reach 15% on average and that Pb had the highest reduction. The infusion had a low extraction power (<10% in mean values) and the extraction reached 20% in Cr and Pb only in the 95th percentile, i.e., the stage with the greatest influence on the final result of the metals found in coffee.

In relation to the review carried out to compare our results with others, it should be noted that few authors have studied the influence of coffee processing on the heavy metal concentration and their conclusions were not comparable. These differences could be due to the metal concentration in the soil, the production parameters and the extraction conditions of the brew, such as extraction time, water temperature, the fineness of the ground coffee, application of pressure, etc. [45,65,66]. The results of Arvay et al. (2018) [53] on Cr concentration were 0.12 mg/kg in green beans and 0.03 mg/kg in roasted beans, which represent a reduction of around 75%. Khunlert et al. (2021) [67] determined that Cr and Pb increased slightly after roasting and had a different extraction percentage with infusion (i.e., 23% of Cr and 1.2% of Pb present in roasted coffee). Adler et al. (2019) [45] found that roasting has different effects on Pb concentration: one sample suffered a reduction of 86%, while in another, Pb increased by 64% after roasting. Concerning metal extraction during coffee infusion, Winiarska-Mieczan et al. (2021) [19] concluded that from 78 to 100% of the Pb passed into the brew. However, Adler et al. (2019) [45] concluded that the extraction of Pb during infusion ranged from 6.8–9.4%. Da Silva et al. (2017) [60] found a mean Pb concentration in roasted beans of 0.75 mg/kg and 0.042 mg/L in the infusion, i.e., an average extraction of approximately 5.6%.

### 3.2. Metal Concentrations in Parchment Coffee Beans

Table 3 gives the mean and standard deviation of metals above the LOD, i.e., iAs, Cr and Pb, analyzed in parchment coffee beans, the multifactor analysis of variance (ANOVA), F-ratio and significant differences for the factors of region and variety. The results show that there are significant differences by region for the three metals.

It should be noted that the values from the San Martín region were higher for all metals, which could be due to the fact that plants absorb metals from soils, which can be contaminated by mining, anthropogenic sources derived from fertilizers, atmospheric deposition and sewage sludge [68]. A study by Arévalo-Gardini et al. (2017) [69] on heavy metal accumulation in cocoa beans conducted in seven regions of Peru concluded that heavy metal concentration tended to be higher in areas close to lakes or rivers and active mining areas such as San Martín or Huánuco. The findings showed that differences in the variety factor were non-significant in iAs (*p*-value 0.7723) and Pb (*p*-value 0.2596) but were significant in Cr (*p*-value 0.0000). These differences may be related to plant-specific biochemical parameters, such as chlorophyll or secondary metabolite concentration, which can influence the accumulation of certain metals [70].

### 3.3. Risk Assessment

The concentration of metals in the cup of coffee was obtained by simulation, taking into account the initial contamination of parchment coffee beans (Section 3.2) and the metal variations due to processing (Table 2). For Cd and Hg, which were not detected, the concentration value used was 2.5E-03, corresponding to LOD/2. Table 4 gives the mean value and distribution percentile obtained for the estimated daily intake and the metrics used to characterize the risk of ingesting metals. The results show that exposure to iAs and Pb was higher by one order of magnitude than for the other metals tested.

For non-genotoxic effects, the HQ was calculated for iAs, Cr, Cd and Hg (see Table 4). In all regions, the 95th percentile values were less than 1, indicating a null probability of undesirable effects. HI was calculated to account for the combined effect and the lowest values were found in Amazonas, Huánuco and San Martín and the highest in Cajamarca and Cusco (around 4.2E-02 and 1.4E-01 in the 50th percentile, respectively), confirming the low risk of coffee consumption. In the same vein, Taghizadeh et al. (2023) [71] concluded that the HI, as the sum of the HQ of Cd, Cr, Cu, Fe and Hg, present in 87 coffee samples collected from retail markets of Iran, was 1E-02. Similarly, a study on roasted coffee beans purchased in Polish markets by Kowalska in 2021 [15] revealed that the HQ for Cd ranged from 6.46E-02 to 9.81E-02, for Hg from 1.1E-03 to 2.6E-03 and for Pb from 4.33E-03 to 2.45E-02. The HI obtained was 1.25E-01, which can be interpreted as unlikely to be associated with adverse health effects. The same conclusion was reached by Winiarska-Mieczan et al. (2021) [19], in another Polish study, finding a low Cd and Pb concentration in the analyzed coffee infusions and the HQ values (6.70E-02 and 2.33E-01 for Pb and Cd, respectively), showing that drinking coffee does not pose a risk for consumers in terms of the concentration of these metals. 

The margins of safety for Pb’s non-genotoxic cardiovascular and nephrotoxicity effects were calculated from the MOE (see Table 4). The mean values obtained for Pb, for the effects considered, were in the order of E+01 in both cases. The POE results confirm the low risk level obtained by the MOE, since in no case did the exposure exceed the reference value (BMDL) [44].

CR was calculated for iAs and Pb. The probability of developing cancer from iAs was considered medium in the 5th and 50th percentiles (between E-04 and E-06) in all regions, although the risk was considered high (>E-04) in the 95th percentile. The Pb risk level was considered low, except for the 95th percentile, with values around E-06. These values are in line with those published by Taghizadeh et al. (2023) [71], who obtained a CR of 3.84E-06 for the presence of iAs in coffee and 2.71E-08 for Pb concentration. Similarly, Kowalska (2021) [15] concluded that the maximum CR for iAs was 1.29-05.

A tornado plot was drawn for the sensitivity analysis of the variance of the inputs to the risk characterization metrics (see Figure 4). In general, the results indicated that consumption, followed by metal concentration in parchment beans and the percentage of metal extraction by infusion, was the most positively influenced parameter, while roasting, body weight and hulling had a negative effect. Focusing on the metrics, consumption had the highest effect in HI (around 90% of the variability). The concentration of metals in parchment beans had a higher influence on the variability in CR of Cr and roasting in CR of iAs.

## 4. Conclusions

This article provides knowledge about the concentration of iAs, Cd, Cr, Hg and Pb in coffee from Peru, considering the factors of the variety and growing area. In addition, it describes a study of the variation of metals during processing until obtaining the coffee infusion, obtaining a predictive model of the different stages, and finally, it analyzes the risk for the consumer due to coffee consumption, opting for a probabilistic view to take into account the variability of the inputs that contribute to the risk and a sensitivity study.

The predictive models obtained showed a reduction in the concentration of metals in all coffee-processing stages, although this varies according to the metal studied. Infusion was the stage with the highest metal reduction, especially iAs (approximately 6% extraction), while the average reduction is lower, around 10% for Cr and 15% for Pb. The results by region for parchment coffee showed that the differences were significant, and that San Martín had the highest values for iAs, Cr and Pb. The study by variety was only significant for Cr, with the Pache variety giving the highest concentrations. Regarding risk, the HI and MOE-POE combination results confirm the low level of risk for the non-genotoxic effects of iAs, Cd, Cr, Hg and Pb. For genotoxic effects, the iAs values, close to E-04 in the 95th percentile, indicated a high risk, while the risk from Pb would be acceptable in this percentile and negligible in the rest.

## Figures and Tables

**Figure 1 foods-12-03254-f001:**
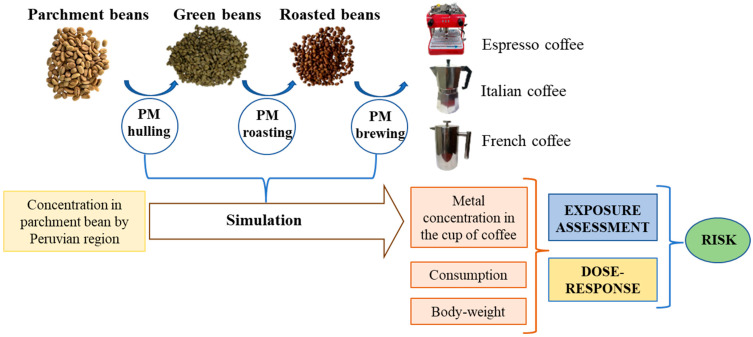
Risk assessment pathway from parchment beans to a cup of coffee.

**Figure 2 foods-12-03254-f002:**
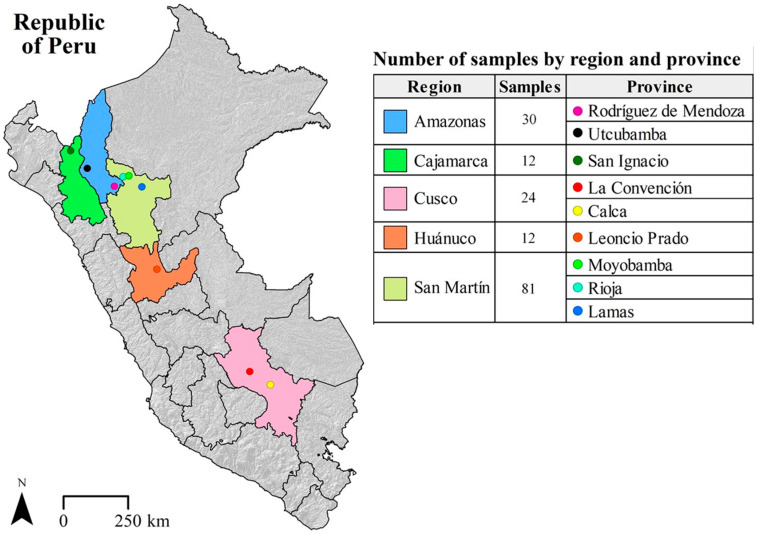
Map of Peru showing the sampling sites.

**Figure 3 foods-12-03254-f003:**
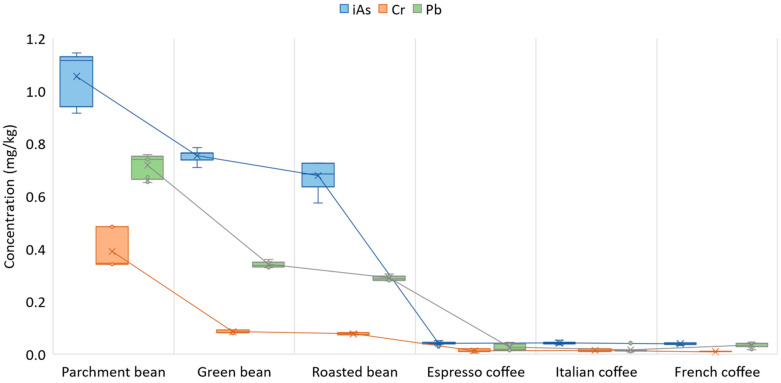
Concentration of iAs, Cr and Pb in parchment, green, roasted beans and coffee infusion obtained by three types of coffee machines: Espresso, French and Italian.

**Figure 4 foods-12-03254-f004:**
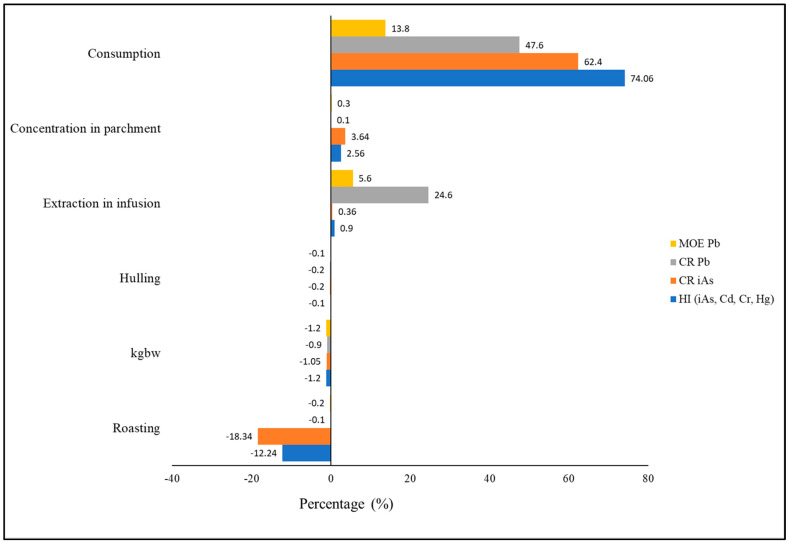
Results of the tornado plot on the analysis of the parameters that most influence variability in the risk characterization metrics.

**Table 1 foods-12-03254-t001:** Validation parameters of the analytical procedure for the determination of iAs, Cd, Cr, Hg and Pb.

Validation Parameters	Heavy Metals				
	iAs	Cd	Cr	Hg	Pb
LOD * (mg/Kg)	0.005	0.005	0.005	0.005	0.005
LOQ ** (mg/Kg)	0.010	0.016	0.010	0.012	0.087
Linearity	0.9962	0.9998	0.9994	0.9835	0.9998
Working range (mg/Kg)	LOQ -4	LOQ -4	LOQ -4	LOQ -5	LOQ -4
Recovery (%)	99	101	98	101	99

* LOD: Limit of detection; ** LOQ: Limit of quantification.

**Table 2 foods-12-03254-t002:** Reduction (%) of the different metals from the parchment bean to the coffee infusion.

Process Stage	iAs			Cr			Pb		
	Mean (SD)	5th	95th	Mean (SD)	5th	95th	Mean (SD)	5th	95th
% Removal in hulling	27.3 (4.4)	18.4	32.3	76.2 (3.2)	71.1	82.9	52.6 (2.8)	48.1	57.1
% Removal in roasting	11.7	4.8	28.4	9.9 (1.5)	7.5	12.4	14.7 (3.7)	8.9	20.5
% Extraction in brewing	5.9 (0.9)	4.8	7.5	13.4 (4.1)	6.8	20.1	8.2 (6.1)	4.0	19.5

**Table 3 foods-12-03254-t003:** Heavy metal concentrations (mean and standard deviation) in parchment coffee beans for five varieties collected from five Peruvian regions. ANOVA F-ratio for region and variety.

Metals	Region					Variety					ANOVA F-Ratio	
	Amazonas	Cajamarca	Cusco	Huánuco	San Martín	Bourbon	Catimor	Caturra	Pache	Típica	Region	Variety
iAs (mg/kg)	0.57 ± 0.21 ^a^	0.50 ± 0.20 ^a^	0.63 ± 0.19 ^a^	0.62 ± 0.09 ^a^	0.94 ± 0.17 ^b^	0.72 ± 0.24 ^a^	0.76 ± 0.24 ^a^	0.76 ± 0.29 ^a^	0.89 ± 0.17 ^a^	0.79 ± 0.24 ^a^	39.425 ***	0.62 ^ns^
Cr (mg/kg)	0.08 ± 0.05 ^a^	0.05 ± 0.02 ^a^	0.12 ± 0.06 ^b^	0.08 ± 0.04 ^a^	0.21 ± 0.12 ^c^	0.11 ± 0.08 ^a^	0.14 ± 0.07 ^ab^	0.17 ± 0.13 ^b^	0.39 ± 0.24 ^c^	0.12 ± 0.09 ^a^	15.317 ***	10.041 ***
Pb (mg/kg)	0.64 ± 0.04 ^a^	0.64 ± 0.03 ^a^	0.67 ± 0.03 ^b^	0.64 ± 0.02 ^a^	0.72 ± 0.05 ^c^	0.68 ± 0.05 ^a^	0.69 ± 0.06 ^ab^	0.69 ± 0.05 ^ab^	0.73 ± 0.04 ^b^	0.67 ± 0.05 ^a^	29.197 ***	1.84 ^ns^

Different letters per row indicate statistically different groups (Tukey test, *p* < 0.05), ns: Not significant; *** *p* < 0.001.

**Table 4 foods-12-03254-t004:** Exposure assessment and risk characterization.

Metal	Region	EDI			HQ				MOE				CR			
		Mean	5th	95th	5th	50th	75th	95th	5th	50th	75th	95th	5th	50th	75th	95th
iAs	Amazonas	4.7E-05	5.1E-06	1.5E-04	1.7E-02	1.0E-01	1.9E-01	4.9E-01	1.3E+01	1.1E+02	2.6E+02	8.2E+02	7.7E-06	4.6E-05	8.7E-05	2.3E-04
	Cajamar	4.4E-05	4.6E-06	1.4E-04	1.5E-02	9.4E-02	1.8E-01	4.5E-01	1.4E+01	1.3E+02	2.7E+02	9.1E+02	6.8E-06	4.2E-05	8.2E-05	2.1E-04
	Cusco	5.3E-05	6.6E-06	1.6E-04	2.1E-02	1.2E-01	2.2E-01	5.5E-01	1.1E+01	1.0E+02	2.2E+02	6.6E+02	9.9E-06	5.2E-05	9.9E-05	2.4E-04
	Huánuco	5.3E-05	7.4E-06	1.6E-04	2.5E-02	1.2E-01	2.2E-01	5.3E-01	1.2E+01	9.9E+01	2.1E+02	5.7E+02	1.1E-05	5.4E-05	9.9E-05	2.4E-04
	S. Martín	8.0E-05	1.1E-05	2.5E-04	3.7E-02	1.8E-01	3.3E-01	8.1E-01	7.5E+00	6.6E+01	1.4E+02	3.9E+02	1.6E-05	8.0E-05	1.5E-04	3.7E-04
Cd	All regions	5.6E-06	9.0E-07	1.6E-05	9.0E-04	3.8E-03	6.8E-03	1.6E-02								
Cr	Amazonas	4.9E-06	4.3E-07	1.6E-05	2.9E-07	1.8E-06	3.7E-06	1.1E-05					2.2E-07	1.4E-06	2.8E-06	8.2E-06
	Cajamar	3.8E-06	3.5E-07	1.2E-05	2.4E-07	1.5E-06	3.0E-06	8.1E-06					1.8E-07	1.1E-06	2.3E-06	6.0E-06
	Cusco	7.9E-06	7.7E-07	2.5E-05	5.1E-07	3.1E-06	6.3E-06	1.7E-05					3.8E-07	2.3E-06	4.8E-06	1.3E-05
	Huánuco	5.4E-06	3.9E-07	1.8E-05	2.6E-07	2.1E-06	4.4E-06	1.2E-05					2.1E-07	1.6E-06	3.3E-06	8.9E-06
	S. Martín	1.4E-05	1.1E-06	4.7E-05	7.0E-07	4.9E-06	1.0E-05	3.1E-05					5.4E-07	3.7E-06	7.8E-06	2.3E-05
Hg	All regions	5.6E-06	9.0E-07	1.6E-05	9.0E-03	3.8E-02	6.8E-02	1.6E-01								
Pb	Amazonas	4.7E-05	5.6E-06	1.6E-04					* 9.5E+00	5.6E+01	1.1E+02	2.7E+02	4.8E-08	2.3E-07	4.5E-07	1.3E-06
	Cajamar	4.7E-05	5.5E-06	1.6E-04					9.6E+00	5.7E+01	1.1E+02	2.7E+02	4.7E-08	2.3E-07	4.5E-07	1.3E-06
	Cusco	5.0E-05	5.9E-06	1.7E-04					8.9E+00	5.3E+01	1.0E+02	2.6E+02	5.0E-08	2.4E-07	4.8E-07	1.4E-06
	Huánuco	4.7E-05	5.6E-06	1.6E-04					9.6E+00	5.6E+01	1.1E+02	2.7E+02	4.8E-08	2.3E-07	4.5E-07	1.3E-06
	S. Martín	5.3E-05	6.2E-06	1.8E-04					8.3E+00	5.0E+01	9.6E+01	2.4E+02	5.3E-08	2.6E-07	5.1E-07	1.5E-06
	Amazonas								** 4.0E+00	2.3E+01	4.5E+01	1.1E+02				
	Cajamar								4.0E+00	2.4E+01	4.6E+01	1.1E+02				
	Cusco								3.7E+00	2.2E+01	4.3E+01	1.1E+02				
	Huánuco								4.0E+00	2.3E+01	4.5E+01	1.1E+02				
	S. Martín								3.5E+00	2.1E+01	4.0E+01	1.0E+02				

* MOE (BMDL_01_) = 1.50E-03 mg/kg-Bw/day; ** MOE (BMDL_10_) = 6.30E-04 mg/kg-Bw/day.

## Data Availability

Raw data can be provided by the corresponding author upon request.

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
