# Peer review of "Probabilistic Risk Characterization of Heavy Metals in Peruvian Coffee: Implications of Variety, Region and Processing"

_foods, 2023, doi:10.3390/foods12173254_

Round 1

Reviewer 1 Report

  1. The main question research is appropriate. The investigation of metals in coffee is really interesting. Human exposure to metals by coffee should be minimal as coffee is voluptuous drink, meaning that the consuption is quite low. However, risk assessment conducted in the study is fine, it all makes good sense. 
  2. Topic of paper could be considered as original. Among foods and beverages, coffee (and particularly the contamination through the process) is not clearly. 
  3. The study has been performed carefully. I stringly believe that the paper could improve knoledges known so far. 
  4. Conclusions and refereces can be considered as appropriate.

Line 50-52 page 2: I suggest to better specify the chemical composition by indicating the percentage of components the authors reported (i.e., potassium, X%; ect). 

Line 53 page 2: I think that it is necessary to specify what “moderate” means.

Line 58, page 2: “Highest concentration” does not make sense. I suggest to indicate average concentrations, to do a comparison and make it clearer to the reader. 

Line 65, page 2: Again. “Some type of cancers” is not precise. 

Line 69, page 2: “(pdf)” should be written (i.e., pdf).

I suggest to write mg/kg as mg kg-1

Line 199-200 page 5: Add units  

Line 207: Add units: 0.1 needs unit. 

English language is quite good. 

Author Response

Response to Reviewer 1 Comments

The authors are grateful for the reviewer's comments.

Point 1: Line 50-52 page 2: I suggest to better specify the chemical composition by indicating the percentage of components the authors reported (i.e., potassium, X%; ect).

Response 1: Given that the concentrations vary significantly according to the type of coffee (instant, macchiato, espresso, origin, type of bean and studies. The authors believe that it is convenient not to give a percentage. However, they recognize the interest in this information and have therefore introduced a new reference corresponding to the USDA that provides a great deal of information on the composition according to the different types of coffee.

Point 2: Line 53 page 2: I think that it is necessary to specify what “moderate” means.

Response 2: Following the references, the authors have specified the term “moderate” meaning 2-5 cups per day.

Point 3: Line 58, page 2: “Highest concentration” does not make sense. I suggest to indicate average concentrations, to do a comparison and make it clearer to the reader

Response 3: The sentence has been rewritten for a better understanding. However, a comparison in percentages between sources (natural and industrialized) was not included since this information is different from the aim of the paper.

Point 4: Line 65, page 2: Again. “Some type of cancers” is not precise.

Response 4: Rewritten with the required precision.

Point 5: Line 69, page 2: “(pdf)” should be written (i.e., pdf).

Response 5: The change has been done.

Point 6: I suggest to write mg/kg as mg kg-1

Response 6: After reviewing the guide for authors, publications of the journal and following the indication of the publication Gutiérrez-Avella & Guardado-Pérez 2010, DOI: 10.1016/S0187-893X(18)30072-7, (https://www.elsevier.es/es-revista-educacion-quimica-78-articulo-formas-expresar-composicion-quimica-el-S0187893X18300727), the authors consider that both units are possible.

Point 7: Line 199-200 page 5: Add units

Response 7: The change has been done.

Point 8: Line 207: Add units: 0.1 needs unit

Response 8: The change has been done.

Reviewer 2 Report

The study analyzes the presence of heavy metals in different stages of the coffee process, including parchment, green and roasted beans, and coffee infusion. The results show that the concentration of heavy metals varies depending on the type of coffee machine used. The study also discusses the potential health risks associated with long-term exposure to heavy metals in coffee. The authors suggest that measures should be taken to reduce the levels of heavy metals in Peruvian coffee and ensure food safety.

Although the material submitted is suitable for this journal, some improvements need to be made.

The validation experiments should be described better. In particular, the method precision results should be presented.

The working range should be defined from the LOQ, not from 0.

The LODs reported for all metals are equal to 0.005, while the LOQs range from 0.010 to 0.087. The authors should explain this difference.

The results presented in Table 1 should be moved to the Results section, not to the Materials and Methods section.

The reference format should be revised. For example, Reference 60 should be written as "Da Silva et al. (2017)".

Author Response

Response to Reviewer 2 Comments

The authors are grateful for the reviewer's comments.

Point 1: The validation experiments should be described better. In particular, the method precision results should be presented.

Response 1: Lines 128-131 have been introduced to complete the description of the validation experiment.

Point 2: he working range should be defined from the LOQ, not from 0.

Response 2: The change has been done.

Point 3: The LODs reported for all metals are equal to 0.005, while the LOQs range from 0.010 to 0.087. The authors should explain this difference.

Response 3: New lines 123-126.

Point 4: The results presented in Table 1 should be moved to the Results section, not to the Materials and Methods section.

Response 4: Although the table is indeed a work carried out by the authors, and that is why the table should be in results, the authors have reviewed articles whose objective is risk, and in most of them, this information is presented in material and methods. Therefore, the authors would also prefer to leave the table in material and methods.

Point 5: The reference format should be revised. For example, Reference 60 should be written as "Da Silva et al. (2017)".

Response 5: All references were reviewed and edited.

Reviewer 3 Report

The paper `present the probabilistic risk characterization of heavy metals in Peruvian coffee and the implications of variety, region and processing of it using the hazard index and cancer risk assessment.

Even though the paper does not provide the novelty the topic is interesting for discussion. The introduction part should include more detail a paragraph that presents clearly the aims, problem statement, contribution, objectives and findings. This is not well elaborated, and it is very difficult to recognise the gap and the contribution need at the first stage. What this paper differs from the cited authors and what brings new to the scientific contribution. How that is relevant to Peru.

Material and methods does not provide very accurate information. The author should explain the background and rationale for the methods chosen, it is refereeing the method used from the paper no. 32 were the authors mostly presented the determination of the heavy metals in lake and sediments. The authors should explain explaining why that method was chosen over their coffee samples.

The use of statistical analysis should be also presented in detail. How this statistics’ chosen in plies to the presentation of findings and what is the accuracy of the results presented.

In the discussion part the authors compared the results with other publication however no clear evidence was provided on the differences.  

The conclusion does not provide the significance pf the paper and does not remain in the main argument.

It can be proofread by a professional lector. 

Author Response

Response to Reviewer 3 Comments

The authors are grateful for the reviewer's comments.

Point 1: The introduction part should include more detail a paragraph that presents clearly the aims, problem statement, contribution, objectives and findings. This is not well elaborated, and it is very difficult to recognise the gap and the contribution need at the first stage. What this paper differs from the cited authors and what brings new to the scientific contribution. How that is relevant to Peru

Response 1: New lines 77-80 try to clarify the contribution of the paper.

Point 2: Material and methods does not provide very accurate information. The author should explain the background and rationale for the methods chosen, it is refereeing the method used from the paper no. 32 were the authors mostly presented the determination of the heavy metals in lake and sediments. The authors should explain explaining why that method was chosen over their coffee samples.

Response 2: The university's laboratory is accredited for determining metals by the method described, which was used in the publication referred to. The coffee beverage, being a liquid sample, has the same procedure as water samples.

Point 3: The use of statistical analysis should be also presented in detail. How this statistics’ chosen in plies to the presentation of findings and what is the accuracy of the results presented

Response 3: The paragraph has been rewritten (new lines 198-201) in order to present more details.

Point 4: In the discussion part the authors compared the results with other publication however no clear evidence was provided on the differences. 

Response 4: The authors only wanted to compare the results, saying which authors obtained more similar values and which got higher or lower results. The main reason for the difference may be the production location, however, factors such as the analytical methods, number of samples, industries around, etc., make it difficult to associate the difference in their results with our results.

Point 5: The conclusion does not provide the significance Of the paper and does not remain in the main argument

Response 5: A new paragraph has been introduced with this aim (new lines 371-377)

Round 2

Reviewer 3 Report

The paper is improved from the first submission.